# Cross-tissue integration of genetic and epigenetic data offers insight into autism spectrum disorder

Shan V. Andrews [1,2], Shannon E. Ellis [3], Kelly M. Bakulski [4], Brooke Sheppard[1,2], Lisa A. Croen [5], Irva Hertz-Picciotto[6,7], Craig J. Newschaffer[8,9], Andrew P. Feinberg[10,11], Dan E. Arking[2,3], Christine Ladd-Acosta [1,2,10] & M. Daniele Fallin[2,10,12]

Integration of emerging epigenetic information with autism spectrum disorder (ASD) genetic results may elucidate functional insights not possible via either type of information in isolation. Here we use the genotype and DNA methylation (DNAm) data from cord blood and peripheral blood to identify SNPs associated with DNA methylation (meQTL lists). Additionally, we use publicly available fetal brain and lung meQTL lists to assess enrichment of ASD GWAS results for tissue-specific meQTLs. ASD-associated SNPs are enriched for fetal brain (OR = 3.55; $P < 0.001$) and peripheral blood meQTLs (OR = 1.58; $P < 0.001$). The CpG targets of ASD meQTLs across cord, blood, and brain tissues are enriched for immune-related pathways, consistent with other expression and DNAm results in ASD, and reveal pathways not implicated by genetic findings. This joint analysis of genotype and DNAm demonstrates the potential of both brain and blood-based DNAm for insights into ASD and psychiatric phenotypes more broadly.

[1] Department of Epidemiology, Johns Hopkins Bloomberg School of Public Health, 615N. Wolfe St, Baltimore, MD 21205, USA. [2] Wendy Klag Center for Autism and Developmental Disabilities, Johns Hopkins Bloomberg School of Public Health, 615 N. Wolfe St, Baltimore, MD 21205, USA. [3] McKusick-Nathans Institute of Genetic Medicine, Johns Hopkins School of Medicine, 733 N. Broadway, Baltimore, MD 21205, USA. [4] Department of Epidemiology, University of Michigan School of Public Health, 1415 Washington Heights, Ann Arbor, MI 48109, USA. [5] Division of Research, Kaiser Permanente Northern California, 2000 Broadway, Oakland, CA 94612, USA. [6] Department of Public Health Sciences, School of Medicine, University of California Davis, 4610 X St, Sacramento, CA 95817, USA. [7] MIND Institute, University of California Davis, 2825 50th St, Sacramento, CA 95817, USA. [8] AJ Drexel Autism Institute, Drexel University, 3020 Market St #560, Philadelphia, PA 19104, USA. [9] Department of Epidemiology and Biostatistics, Drexel University Dornsife School of Public Health, 3125 Market St, Philadelphia, PA 19104, USA. [10] Center for Epigenetics, Institute for Basic Biomedical Sciences, Johns Hopkins School of Medicine, 733 N. Broadway, Baltimore, MD 21205, USA. [11] Department of Medicine, Johns Hopkins School of Medicine, 733 N. Broadway, Baltimore, MD 21205, USA. [12] Department of Mental Health, Johns Hopkins Bloomberg School of Public Health, 624 N. Broadway, Baltimore, MD 21205, USA. Correspondence and requests for materials should be addressed to C.L.-A. (email: claddac1@jhu.edu) or to M.D.F. (email: dfallin@jhu.edu)

Autism spectrum disorder (ASD) is a complex neurodevelopmental disorder characterized by deficits in social communication and interaction as well as restricted repetitive behavior[1]. ASD has a strong genetic basis[2, 3], and most findings to date have been rare variants, including inherited and de novo mutations as well as copy number variations[4–6]. Although rare variants explain a relatively small proportion of all ASD cases[7], they provide converging evidence for three key biological processes implicated in ASD, including epigenetic regulation[6, 8, 9]. Other lines of evidence also implicate epigenetic mechanisms in ASD[10–14]. Common genetic variation also plays a role, similar to other complex psychiatric diseases[15–17], but mega-analysis GWAS results from the Psychiatric Genomics Consortium ASD workgroup (PGC-AUT)[18] are only recently available, and thorough examination of the biology implicated by common variants has not yet been fully pursued. Previous studies of neuropsychiatric disorders[19–21] have demonstrated the enrichment of GWAS results for expression quantitative trait loci (eQTLs), providing insights into the functional biology of discovered GWAS SNPs, assuming those SNPs confer some risk through regulatory mechanisms. Given the implications of epigenetic regulation in ASD, a similar approach exploring enrichment of SNPs controlling epigenetic marks, such as DNA methylation, may be fruitful, assuming similarly that ASD genetic risk may act in part through epigenetic regulation. As with expression loci, genetic variation contributes to DNAm levels locally and distally[22, 23] and thus integration of methylation QTLs (meQTLs), or SNPs that are highly associated with DNAm, and autism-associated GWAS results may inform our understanding of autism GWAS findings. Moreover, meQTLs are enriched in top hits for bipolar disorder[19] and schizophrenia[20, 22], which have well-established genetic overlap with ASD[16].

Epigenetic patterns necessarily differ across tissue types, given their role in cell differentiation and expression. For brain-related conditions like ASD, careful consideration of tissue source for epigenetic analyses is warranted[24]. This must be balanced by consideration of tissue availability, which is limited for ASD brain tissue. We and others have shown blood-based epigenetic biomarkers are useful in psychiatric conditions, including ASD[25, 26], while recognizing the limitations and need for comparison to the brain-derived data wherever possible[24, 27, 28]. ASD-related epigenetic differences have also been observed in buccal[29], lymphoblastoid cell line[30], and postmortem brain samples[31–33], as well as in the sperm from fathers of children with ASD[34].

Blood–brain DNAm concordance studies have not frequently observed high correlation of DNAm levels at specific sites across tissues; however, when such concordance is observed, it is likely due to genetic influences[27, 28]. meQTL signals overlap in adult brain and blood tissues[35], suggesting blood-derived meQTLs may also reflect SNP–DNAm relationships in brain tissue, though this relationship has rarely been tested.

This study used lists and locations of meQTLs and their CpG targets, termed meQTL maps, from cord blood, peripheral blood, and fetal brain tissues to characterize and prioritize ASD GWAS SNPs and the CpG sites under their control. We created such meQTL maps from our own data for infant cord and childhood peripheral blood tissues, and used publicly available meQTL maps for brain and lung tissues to examine cross-tissue meQTLs. We find that ASD GWAS signals are enriched for meQTLs in peripheral blood and fetal brain. The CpG site targets controlled by ASD-associated SNPs are enriched for immune response pathways, and can implicate genes not directly identified by GWAS results alone. Finally, we extend the characterization of SNP-controlled CpG sites to neuropsychiatric disease more generally, and discover their enrichment for specific regulatory features within and across tissue type. Our work demonstrates the utility of jointly analyzing the GWAS and DNAm data for insights into ASD and neuropsychiatric disease.

## Results

**Creating meQTL maps.** We identified meQTL SNPs using combined GWAS and 450 K methylation array data available on both peripheral blood and cord blood samples. For these analyses, we defined study-specific parameters that were optimal for each data set and determined the $P$ value (Wald test) to control the false discovery rate (FDR) at 10, 5, and 1%. In peripheral blood, we identified 1,878,577 meQTLs controlling DNAm at 85,250 CpGs; in cord blood, we found 1,252,498 meQTLs controlling DNAm at 35,905 CpGs, both at FDR = 5%. Peripheral blood and cord blood meQTLs, on average, were associated with 4.83 and 2.56 CpG sites, respectively. Statistical significance was inversely related to distance between SNP and CpG site (Supplementary Fig. 1). We have provided a full list of all identified peripheral and cord blood meQTLs and their associated CpG sites at FDR = 5% (Supplementary Data 1 and 2).

We used publicly available lung[23] (to include a likely non ASD-related tissue) and fetal brain[22] meQTL lists and the $P$ value cutoffs stated in those respective studies (Wald $P$ = 1e–08 for fetal

**Table 1 Descriptive characteristics, meQTL query parameters, and meQTL summary results for four tissues examined**

| | Sample size | Meth SD cutoff[a] | SNP MAF threshold[b] | Max SNP to CpG distance[c] | meQTL $P$ value thresholds[d] | # of meQTLs identified | # of meQTL targets identified |
|---|---|---|---|---|---|---|---|
| Fetal brain[e] | 166 | NA | 5% | 1 Mb | 1.0e-08[f] | 299,992[f] | 7863[f] |
| Peripheral blood | 339 | 0.15 | 2.75% | 1 Mb | 3.1e-05[g] | 2,003,443[g] | 95,195[g] |
| | | | | | 1.0e-05[h] | 1,878,577[h] | 85,250[h] |
| | | | | | 3.0e-07[i] | 1,598,033[i] | 68,860[i] |
| Cord blood | 121 | 0.15 | 7% | 500 Kb | 8.5e-06[g] | 1,374,554[g] | 41,681[g] |
| | | | | | 2.7e-06[h] | 1,252,498[h] | 35,905[h] |
| | | | | | 2.0e-07[i] | 1,032,370[i] | 28,423[i] |
| Lung[e] | 210 | NA | 3% | 500 Kb | 4.0e-05[h] | 22,866[h] | 34,304[h] |

[a]The probe standard deviation across samples that was used as an inclusion criterion for CpG sites in the meQTL query (blood data sets only)
[b]The MAF cutoff used as an inclusion criterion for SNPs in the meQTL query
[c]The maximum distance between the SNP and CpG site used in the meQTL query for the peripheral blood, cord blood, and lung data sets, and the value at which results for filtered in the fetal brain results
[d]SNP-to-CpG association $P$ values considered in subsequent analyses
[e]Publicly available data
[f]FDR not specified
[g]FDR = 10%
[h]FDR = 5%
[i]FDR = 1%

**Table 2 Enrichment statistics for meQTLs derived from 4 tissue types in ASD GWAS SNPs**

| | ASD *P* value = 1e–03 | | | ASD *P* value = 1e–04 | | |
| | meQTL *P*-value = 1e–08 | | | meQTL *P* value = 1e–08 | | |
|---|---|---|---|---|---|---|
| Fetal brain[a] | 1.70 (<0.001) | | | 3.55 (<0.001) | | |
| | **meQTL FDR = 10%** | **meQTL FDR = 5%** | **meQTL FDR = 1%** | **meQTL FDR = 10%** | **meQTL FDR = 5%** | **meQTL FDR = 1%** |
| Peripheral blood[b] | 1.22 (<0.001) | 1.20 (<0.001) | 1.23 (<0.001) | 1.31 (0.001) | 1.40 (<0.001) | 1.58 (<0.001) |
| Cord blood[b] | 1.14 (0.032) | 1.21 (0.011) | 1.20 (0.023) | 1.13 (0.299) | 1.10 (0.392) | 1.10 (0.406) |
| Lung[a] | — | 1.09 (0.343) | — | — | 0.80 (0.301) | — |

Enrichment fold statistics and *P* values based on 1000 permutations
[a]LD pruning performed with 1000 Genomes CEU samples
[b]LD pruning performed with the study-specific genotype data. See Methods for additional details

**Table 3 Gene Ontology terms significantly enriched in multiple tissue types in comparison of ASD-related meQTL targets to meQTL targets generally**

| Term | Peripheral blood scaled rank[a] | Cord blood scaled rank[a] | Fetal brain scaled rank[a] |
|---|---|---|---|
| Response to interferon-gamma | 0.14 | 0.11 | 0.11 |
| Positive regulation of relaxation of cardiac muscle | 0.20 | 0.46 | 0.30 |
| Production of molecular mediator of immune response | 0.65 | 0.22 | 0.28 |
| Cellular response to interferon-gamma | NA | 0.07 | 0.09 |
| Detection of bacterium | NA | 0.18 | 0.06 |
| Detection of biotic stimulus | NA | 0.26 | 0.04 |
| T-helper 1 type immune response | NA | 0.08 | 0.34 |
| Regulation of interleukin-10 secretion | NA | 0.09 | 0.43 |
| Interferon-gamma production | NA | 0.57 | 0.19 |
| Regulation of interleukin-4 production | NA | 0.24 | 0.62 |
| Interleukin-4 production | NA | 0.29 | 0.60 |
| Interleukin-10 production | NA | 0.25 | 0.74 |
| Tongue development | NA | 0.68 | 0.32 |
| Inflammatory response to antigenic stimulus | NA | 0.32 | 0.81 |
| Endochondral bone growth | NA | 0.71 | 0.53 |
| Antigen processing and presentation of peptide or polysaccharide antigen via MHC class II | 0.01 | 0.05 | NA |
| T-cell costimulation | 0.05 | 0.01 | NA |
| Positive regulation of hormone secretion | 0.09 | 0.04 | NA |
| Antigen receptor-mediated signaling pathway | 0.08 | 0.13 | NA |
| Immunoglobulin production involved in immunoglobulin mediated immune response | 0.24 | 0.03 | NA |
| Single organismal cell-cell adhesion | 0.23 | 0.12 | NA |
| Single organism cell adhesion | 0.34 | 0.16 | NA |
| Negative regulation of nonmotile primary cilium assembly | 0.16 | 0.39 | NA |
| Antigen processing and presentation of polysaccharide antigen via MHC class II | 0.02 | 0.58 | NA |
| Polysaccharide assembly with MHC class II protein complex | 0.03 | 0.59 | NA |
| Protein-carbohydrate complex subunit organization | 0.04 | 0.61 | NA |
| Microtubule sliding | 0.29 | 0.38 | NA |
| MHC protein complex assembly | 0.06 | 0.75 | NA |
| Negative regulation of serine-type peptidase activity | 0.41 | 0.41 | NA |
| Regulation of satellite cell activation involved in skeletal muscle regeneration | 0.39 | 0.45 | NA |
| Protein repair | 0.43 | 0.43 | NA |
| Regulation of serine-type peptidase activity | 0.48 | 0.47 | NA |
| Protein localization to basolateral plasma membrane | 0.46 | 0.55 | NA |
| Lymphocyte migration into lymphoid organs | 0.47 | 0.62 | NA |
| Peyer's patch morphogenesis | 0.60 | 0.70 | NA |
| Regulation of homeostatic process | 0.45 | 0.92 | NA |
| Skeletal muscle satellite cell activation | 0.88 | 0.63 | NA |

[a]Scaled rank refers to enrichment *P* value-based rank divided by the number of marginally significant terms post REVIGO filtering for that tissue (peripheral blood: 95, cord blood: 76, fetal brain: 47)
'NA' shown for terms that did not appear in these lists for that tissue. Terms are lumped into sections based on cross-tissue overlap: Section 1—all three tissues, Section 2—cord blood and fetal brain, Section 3—peripheral blood and cord blood. Within each of these sections, terms are arranged according to the sum of the scaled ranks. See Methods for additional details

brain and Wald $P = 4e{-}05 = $ FDR 5% for lung). In fetal brain, there were a total of 299,992 meQTLs controlling 7863 CpGs, and in lung there were 22,866 meQTLs controlling 34,304 CpG sites. The data set characteristics, meQTL parameters, and $P$ values used are summarized in Table 1. In all tissues, meQTL targets (CpG sites controlled by meQTLs) implicate additional genes that are not accounted for by their corresponding meQTLs (Supplementary Table 1).

There were 2,704,013 overlapping SNPs considered for meQTL discovery across peripheral blood, cord blood, and fetal brain analyses. Of these, 125,869 (4.65%) were identified as meQTLs in all three tissues, 407,722 (15.08%) were meQTLs only in peripheral and cord blood, 30,691 (1.14%) were meQTLs only in peripheral blood and fetal brain, and 528 (0.02%) were meQTLs only in cord blood and fetal brain (Supplementary Table 2).

**Enrichment of meQTLs in ASD GWAS SNPs across 4 tissue types.** We observed enrichment of fetal brain meQTLs at both the more liberal GWAS SNP $P$ value threshold (enrichment fold = 1.70, permutation $P_{enrichment} < 1e{-}03$ at Wald $P_{GWAS} < 1e{-}03$), and at a more stringent GWAS $P$ value threshold (3.55, permutation $P_{enrichment} < 1e{-}03$ at Wald $P_{GWAS} < 1e{-}04$) (Table 2). There was no association with lung meQTLs at either the more liberal (1.09, permutation $P_{enrichment} = 0.343$) or more stringent (0.80, permutation $P_{enrichment} = 0.301$) threshold.

In peripheral and cord blood, we considered multiple GWAS SNP $P$ value thresholds as well as multiple meQTL discovery thresholds (the latter not available in the brain and lung public data). There was significant meQTL enrichment for all GWAS and meQTL thresholds considered using peripheral blood meQTLs (enrichment fold range = 1.20–1.58, permutation $P_{enrichment} < 1e{-}03$; Table 2). However, in cord blood, meQTL enrichment was only observed for a liberal GWAS SNP threshold (range = 1.14–1.21, permutation $P_{enrichment} = 0.011$–0.032 at Wald $P_{GWAS} < 1e{-}03$). This was not statistically significant after considering a Bonferonni correction to account for the 16 enrichment tests performed.

**Gene ontology enrichment analyses of meQTL targets.** We examined the biological functions of meQTL targets of ASD SNPs specifically compared to meQTL targets generally. We identified 210, 66, and 53 meQTL targets associated with ASD SNPs in peripheral blood, cord blood, and fetal brain, respectively. After mapping these CpG sites to genes, performing GO enrichment analyses, and removing overlapping GO terms, there were a total 95, 76, and 47 nominally significant (hypergeometric test $P < 0.05$) biological processes, respectively.

A total of 37 biological processes were present across either two or three tissues (Table 3, Supplementary Data 3–5), many of them relating to immune system function. Of these, three terms overlapped across all three tissues, 12 processes were enriched in cord blood and fetal brain but not peripheral blood, and 22 processes were present in both the peripheral and cord blood but not in fetal brain.

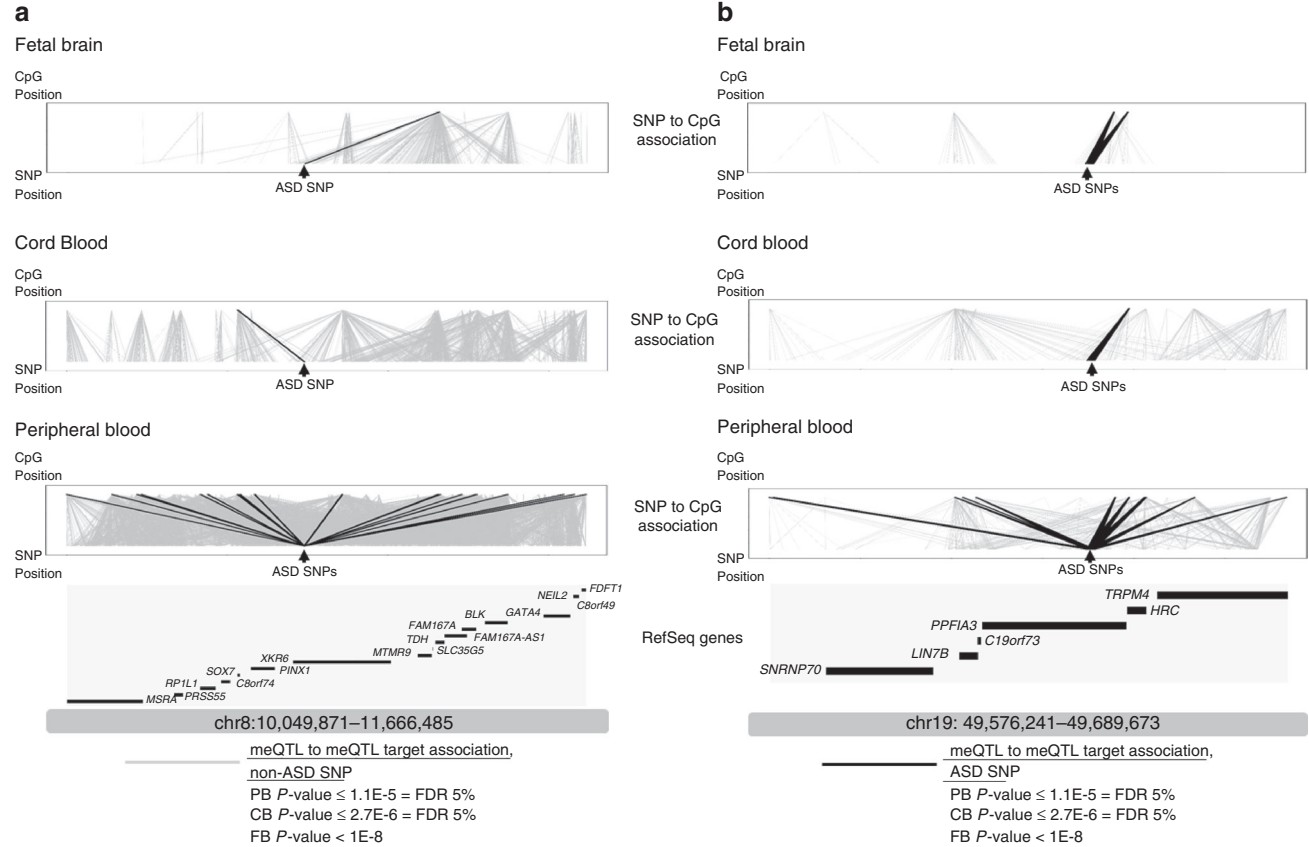

**Fig. 1** 'Expansion' of ASD loci through meQTL mapping in peripheral blood, cord blood, and fetal brain. Each tissue-specific panel presents, from bottom to top: genomic location, gene annotations, SNP locations, SNP–CpG associations, CpG locations. *Light gray* meQTL association lines denote all SNP to CpG associations in that tissue type; *Dark* meQTL association lines denote SNP–CpG associations for ASD-associated SNPs in PGC ($P$ value <= 1e–04). **a** Locus at chr8; **b** Locus at chr19. Data are presented for meQTL maps for fetal brain (*top*); cord blood meQTLs (*middle*), and peripheral blood meQTLs (*bottom*). Please note locus coordinates differ from those in Supplementary Data 6 because in this context they encompass the locations of meQTL target CpG sites

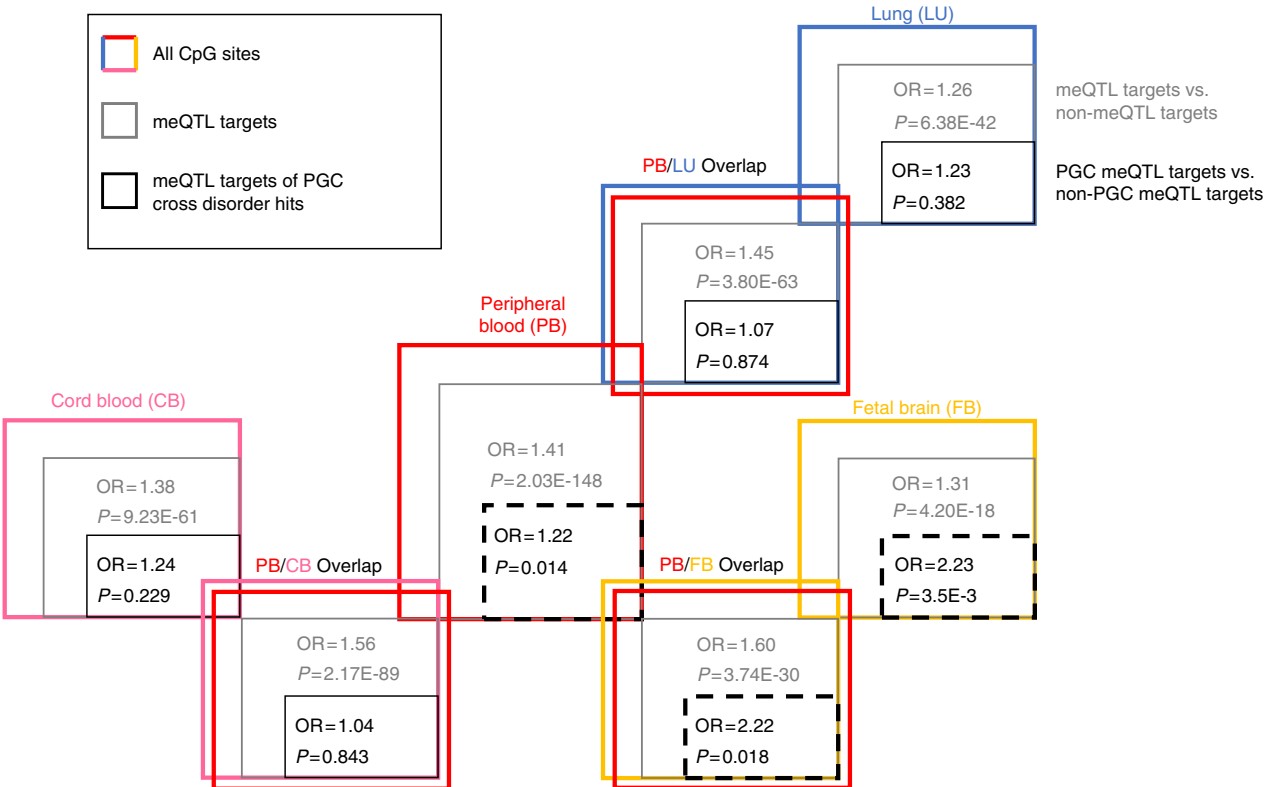

**Fig. 2** Enrichment of meQTL target CpG sites in DNaseI hypersensitive sites. We identified the meQTL targets (at FDR 5% in peripheral blood, cord blood and lung, and past 1e−08 *P* value threshold in fetal brain results) in peripheral blood, cord blood, fetal brain, and lung as well those meQTL targets that were present in the overlap of peripheral blood with the other three tissues. Odds ratio and *P* value in *gray* text represent enrichment fold statistic and *P* value from Fisher's exact tests examining the tendency of meQTL targets to overlap with DHS sites compared to CpG sites that were not meQTL targets. Odds ratio and *P* value in *black* text represent enrichment fold statistic and *P* value from Fisher's exact tests examining the tendency of meQTL targets of significant (*P* value <= 1e−04) SNPs from the PGC cross-disorder results or their proxies ($r^2 >= 0.8$) to overlap with DHS sites compared to CpG sites that were not meQTL targets of the same SNPs. A full list of enrichment statistics and *P* value for both tests against a total of 181 cell-type-specific DHS sites, cell-type-specific chromatin marks, and transcription factor-binding sites is available in Supplementary Data 7 and 8

To test whether our findings were unique to ASD meQTL targets, we performed the same analysis comparing all meQTL targets to all CpG sites. (Supplementary Figs. 2–4). Though there were some immune-related pathways discovered for fetal brain ASD meQTL targets that are also enriched in meQTLs generally, this was not the case in peripheral and cord blood.

**Expansion of ASD GWAS loci via ASD meQTL target locations**. The location of CpG targets for particular meQTL associations can further elucidate genes or regions relevant to ASD risk beyond the genomic location of the associated SNP variant. Of the 1094 ASD-associated PGC SNPs (Wald *P* < 1e−04), five (0.46%) were detected as meQTLs across peripheral blood, cord blood, and fetal brain tissues (Supplementary Table 3, Supplementary Data 6). Consideration of the CpG DNAm targets of these SNPs implicates genes not directly annotated to the SNPs themselves. For example, ASD SNPs in *XKR6* target CpGs in *TDH* in both peripheral blood and fetal brain, and target CpGs in *SOX7* in peripheral blood and cord blood (Fig. 1a). A similar result can be seen for ASD SNPs in *PPFIA3* with meQTL target CpGs that implicate *HRC* (Fig. 1b).

**Characterizing meQTL targets for regulatory feature overlap**. We sought to quantify the propensity of regulatory features to overlap with meQTL targets within and across tissue type, and particularly whether meQTL targets of SNPs associated with psychiatric conditions have specific regulatory features. Individual and overlapping tissue meQTL target lists were compared for regulatory feature annotation. First, among psychiatric condition-associated SNPs (via the PGC cross-disorder analysis[17]), their meQTL targets were significantly enriched for DNaseI hypersensitive sites (DHSs) in peripheral blood (OR = 1.22, Fisher's exact *P* = 0.014), fetal brain (OR = 2.23, Fisher's exact *P* = 3.5e−03), and peripheral blood-fetal brain overlap lists (OR = 2.22, Fisher's exact *P* = 0.018; *black* font and boxes, Fig. 2), compared to meQTL targets of SNPs not associated with psychiatric conditions. Further, there was marginally significant enrichment of CD14 cell-specific DHSs (OR = 2.42, Fisher's exact *P* = 0.013; Supplementary Data 7) in the peripheral blood-fetal brain list. Few chromatin marks met Bonferroni significance (Fisher's exact *P* ≤ 3.95e−05) defined by the 181 tests of regulatory features performed in all 7 lists of meQTL targets, though numerous marginally significant enrichment associations were observed for blood H3K36me3 (active) and blood H3K27me3 (repressive). Transcription factor-binding sites (TFBSs) with observed enrichment include (Supplementary Data 7) STAT1 for fetal brain (OR = 4.32, Fisher's exact *P* = 2.66e−05) and peripheral blood (OR = 2.24, Fisher's exact *P* = 3.56e−08), TAF1 for peripheral blood (OR = 1.53, Fisher's exact *P* = 2.24e−06), cord blood (OR = 2.24, Fisher's exact *P* = 4.01e−06), and fetal brain (OR = 3.2, Fisher's exact *P* = 4.40e−06), and POL2RA for peripheral blood (OR = 1.38, Fisher's exact *P* = 1.14e−06), cord blood (OR = 2.28, Fisher's exact *P* = 3.54e−08), and their overlap (OR = 2.20, Fisher's exact *P* = 9.63e−09).

When considering meQTL targets generally (regardless of their status of being downstream of PGC SNPs), compared to non-meQTL-target CpGs, enrichment was observed for DHSs for all seven meQTL target lists, with the largest effect sizes among the peripheral blood-cord blood overlap list and the peripheral blood-fetal brain overlap list (gray font and boxes, Fig. 2). In fact, a large number of regulatory features were significantly enriched among meQTL targets for each tissue, including lung, given the very large sample size of CpG sites (Supplementary Data 8). However, these were typically small effects. Larger, but still moderate enrichment effects were primarily seen for cross-tissue overlap meQTL lists, particularly for the peripheral blood-cord blood overlap list and the peripheral blood–fetal brain overlap list.

## Discussion

This study integrating ASD GWAS results and meQTL maps provides insights about ASD etiology using data within and across tissue types. First, using blood samples from birth and early life, we identify meQTL maps and compare them to previously reported fetal brain tissue meQTLs, showing a subset of SNPs that are meQTLs across all three tissues., The highest percent overlap is between peripheral and cord blood meQTL maps, as is expected given their tissue similarity. When examining enrichment among ASD GWAS results, we observe enrichment of peripheral blood ($1.20 \leq OR \leq 1.58$; permutation $P < 0.001$) and fetal brain ($OR = 1.70$ and $3.55$; permutation $P < 0.001$) meQTLs. When considering the biological processes annotated to ASD meQTL targets, we see enrichment for immune-related pathways using all three tissue meQTL maps. Further, specific ASD meQTL targets may suggest regions for functional follow-up of ASD genetic associations. Finally, we identify several regulatory elements that preferentially overlap with meQTL targets associated with known SNPs for neuropsychiatric disease generally. Our results demonstrate the utility of meQTLs and their CpG targets for insights into ASD and neuropsychiatric disease overall.

Comparison of meQTL lists across tissues presents several challenges for interpretation of results. First, each set of samples came from a different study source, reflecting different sets of individuals and different sampling strategies, as well as differences in sample size and in genotyping and methylation array platforms. For example, we expected and observed considerable overlap between cord and peripheral blood meQTL signals, and less overlap with brain. The lack of further cross-tissue concordance with brain and blood could be due to differential statistical power between studies, lack of SNP or CpG overlap on arrays, or differences in pipelines used for meQTL discovery (choices of window size, SNP minor allele frequency (MAF), etc). In our functional characterization of meQTL targets, we used down sampling of peripheral blood results to make them comparable to the sample size and meQTL query pipeline decisions of the tissue to which it was being compared. While this is likely an incomplete solution, it is a step toward harmonization that has not been carried out in other studies.

We demonstrate that the joint analysis of SNP and DNAm data can reveal insights towards ASD etiology not apparent when looking at either type of the data alone. It is important to examine the biological implications of the genes implicated by SNPs, as well as the genes and regulatory functions implicated by DNAm. When considering the ASD SNPs, we found enrichment of fetal brain and peripheral blood meQTLs. The enrichment was stronger at increasingly stringent meQTL P value and ASD P value thresholds, bolstering confidence in these findings. These results are also concordant with similar studies of schizophrenia,

a disorder with known genetic overlap to ASD[16], that have demonstrated enrichment in fetal brain meQTLs[22] and peripheral blood meQTLs[20]. A previous study examining enrichment of eQTLs in ASD, GWAS SNPs observed enrichment in parietal and cerebellar eQTLs but not lymphoblastoid cell line eQTLs[21], though the GWAS results in that report likely differ greatly from those of the larger PGC-AUT mega-analysis. Crucially, we did not observe enrichment of lung meQTLs, supporting the specificity of fetal brain and peripheral blood results. However, we also did not observe an enrichment of cord blood meQTLs, suggesting the role of ASD-related DNAm marks in peripheral tissues may be developmentally regulated or a function of age.

Additional insights may be gained through examination of specific CpG targets of the ASD-related SNPs. Among CpG sites that are targets of ASD SNP meQTLs, there is an abundance of immune response-related pathways, using brain, peripheral blood, or cord blood meQTL lists. This immune enrichment was not seen when considering CpG targets of all meQTLs in blood (not just the ASD SNPs), suggesting specificity to ASD. However, such enrichment was seen for all meQTL targets in fetal brain. This may be a consequence of the complications during pregnancy that resulted in fetal tissue collection (56–166 days post conception[22]). Though many immune-related disorders are known to be comorbid with ASD[36], previous enrichment-type analysis for genetic variants alone have not highlighted immune-related pathways, instead implicating chromatin regulation, synaptic function, and Wnt signaling[6, 9], particularly for genes implicated via rare variants. However, several gene expression and epigenetic studies of ASD have implicated immune function in both brain tissue[31, 37–39] and peripheral blood[40, 41]. Our results are concordant with these expression and epigenetic studies but still suggest a role for genetic variation in contributing to immune dysregulation in ASD, through SNP control of DNAm.

Beyond genome and epigenome enrichment analyses, specific meQTL targets also helped to "expand" ASD GWAS-implicated regions to include CpG sites, and their associated genes. While this does not increase or decrease statistical support for a particular GWAS SNP finding, better characterization of the functional architecture of the region can inform follow-up analyses of these hits. Two GWAS loci displayed evidence of meQTLs in peripheral blood, cord blood, and fetal brain, and many more loci displayed evidence of meQTLs in at least one tissue. These target CpG sites, and the genes they implicate, would not be identified via traditional genetic (i.e., GWAS) analyses, since the sequence itself does not show ASD-related variability in these areas. Insights emerge only through the integration of the SNP and DNAm data. Current PGC-AUT GWAS results are likely underpowered to provide reliable genome-wide hits. As larger GWAS of ASD emerge with higher-confidence findings, this cross-tissue meQTL mapping approach should be used to expand regions for follow-up, as recently demonstrated for schizophrenia in fetal brain[22].

Finally, we sought to understand the propensity of meQTL targets, both generally and those controlled by psychiatric disorder-related SNPs, to overlap with regions of known functional activity. MeQTL targets of psychiatric SNPs in peripheral blood, fetal brain, and their intersection significantly overlapped with DHS sites, a result that is concordant with our observation of meQTL enrichment among ASD SNPs limited to peripheral blood and fetal brain. We also identified specific TFBSs enriched in psychiatric disorder meQTL targets such as TAF1 and STAT1. Recently, a study of nine families demonstrated both de novo and maternally inherited single nucleotide changes in TAF1 to be associated with intellectual disability, facial dysmorphology, and neurological manifestations[42]. Our finding that binding sites for the TAF1 transcription factor overlap meQTL targets of

**Table 4 Samples downloaded from Roadmap Epigenomics Project for 5 different histone modifications**

|  | H3K27me3 | H3K36me3 | H3K4me1 | H3K4me3 | H3K9me3 |
|---|---|---|---|---|---|
| Adult lung | NA | GSM1059437 | GSM1059443 | GSM1227065 | GSM1120355 |
|  | GSM1220283 | GSM956014 | GSM910572 | GSM915336 | GSM906411 |
| Fetal brain | GSM621393 | GSM621410 | GSM706850 | GSM621457 | GSM621427 |
|  | GSM916061 |  |  |  | GSM916054 |
| Peripheral blood | GSM1127130 | GSM1127131 | GSM1127143 | GSM1127126 | GSM1127133 |
|  | GSM1127142 | GSM613880 |  |  | GSM613878 |

psychiatric SNPs could serve a basis for future functional studies examining the link between *TAF1* mutations and adverse neurological phenotypes. Lastly, mutations in *STAT1* have been linked to early life combined immunodeficiency[43]. The significant overlap with STAT1 TFBSs could thus serve as a starting point for functional work looking to understand the role of immune disorders in ASD and psychiatric phenotypes generally.

We also considered regulatory feature overlap with meQTL targets in general, for comparison to results for psychiatric meQTLs. Given the very large number of meQTL targets when not restricting to those downstream of psychiatric GWAS SNPs, many cell type-specific DHS sites and chromatin marks showed significant enrichment for meQTLs in general. Among within-tissue analyses, the effect sizes were small. However, enrichment seen in meQTL targets that overlapped peripheral blood and cord blood, and meQTL targets that overlapped peripheral blood and fetal brain was of larger effect size (though still moderate). This is consistent with the idea that genetically controlled CpGs are more likely to have common function across tissues (given they all share genomic sequence) compared to those less under direct genetic control. For example, one study has demonstrated that cross-tissue meQTLs (the SNPs, rather than meQTL CpG targets) are enriched for miRNA-binding sites[35]. We have not found literature examining cross-tissue meQTL targets themselves. This finding suggests an important area for future research, that could give greater context to our and future investigations of cross-tissue meQTL targets in a disease context specifically.

In summary, in our work, we perform a genome-wide study of meQTLs in the context of ASD. The results point to the utility of both brain and blood tissues in studies of ASD that integrate the epigenetic data to enhance current GWAS findings for ASD. We show the utility of examining the meQTL targets of ASD SNPs in providing insights into functional roles like immune system processes that would not be apparent via genotype-based analysis in isolation. Our work suggests that genetic and epigenetic data integration, from a variety of tissues, will continue to provide ASD-related functional insights as GWAS findings and meQTL mapping across a variety of tissues improve.

## Methods

**Cord blood samples.** Cord blood DNA was obtained from newborn participants of the Early autism risk longitudinal risk Investigation (EARLI), an enriched-risk prospective birth cohort described in detail elsewhere[44]. The EARLI study was approved by Human Subjects Institutional Review Boards (IRBs) from each of the four study sites (Johns Hopkins University, Drexel University, University of California Davis, and Kaiser Permanente). Informed consent was obtained from all participating families. The 232 mothers with a subsequent child born through this study had births between November 2009 and March 2012. Infants were followed with extensive neurophenotyping until age three, including ASD diagnostics.

**Cord blood DNA methylation.** Cord blood biospecimens were collected and archived at 175 births. DNA was extracted using the DNA Midi kit (Qiagen, Valencia, CA) and samples were bisulfite treated and cleaned using the EZ DNA methylation gold kit (Zymo Research, Irvine, CA). DNA was plated randomly and assayed on the Infinium HumanMethylation450 BeadChip (Illumina, San Diego, CA), or "450k", at the Center for Inherited Disease Research (CIDR, Johns

Hopkins University). Methylation control gradients and between-plate repeated tissue controls ($n = 68$) were used[34].

The minfi library (version 1.12)[45] in R (version 3.1) was used to process raw Illumina image files with the background correcting and dye-bias equalization method: normal-exponential using out-of-band probe (Noob)[46, 47]. Probes with failed detection $P$ value ($> 0.01$) in $>10\%$ of samples were removed ($n = 661$), as were probes annotated as cross-reactive ($n = 29,233$)[48] and those mapping to sex chromosomes ($n = 11,648$). All cord samples passed sample-based filters (sex matching, detection $P$ values $> 0.01$ in greater than 1% of sites). The pre-processed data were adjusted for batch effects related to the hybridization date and array position using the ComBat() function[49] in the sva R package (version 3.9.1)[50]. The methylation data were available from 175 cord blood samples at 445,241 probes.

**Cord blood genotyping.** Overlapping cord blood DNA methylation and the corresponding SNP data were available on 171 EARLI cord blood samples. The genotype data were generated for 841 EARLI family biosamples and 18 HapMap control samples run on the Omni5 plus exome (Illumina, San Diego, CA) genotyping array at CIDR (Johns Hopkins University), generating the data on 4,641,218 SNPs. The duplicated HapMap sample concordance rate was 99.72% and the concordance rate among five EARLI samples with blind duplicates was 99.9%. Samples were removed if they were HapMap controls ($n = 18$), technical duplicates ($n = 5$; selected by frequency of missing genotypes), or re-enrolled families/other relatedness errors ($n = 9$). No samples met the following additional criteria for exclusion: missing genotypes at $> 3\%$ of probes, or excess heterozygosity or homozygosity (4 SD). Probes were removed for CIDR technical problems ($n = 94,712$), missing genomic location information ($n = 8124$). Among probes with high minor allele frequencies ($> 5\%$), SNPs with a missing rate $> 5\%$ were excluded ($n = 8902$) and among probes with low minor allele frequencies ($< 5\%$) SNPs with a missing rate $> 1\%$ were excluded ($n = 65,855$). There were 827 samples and 4,463,625 probes at this stage and SNPs out of Hardy–Weinberg equilibrium ($\chi^2$ test $P < 10^{-7}$) were flagged ($n = 2170$). Samples were merged with the 1000 genomes project (1000GP, version 5) data[51] and EARLI ancestries were projected into four categories (White, Black, Asian, Hispanic). EARLI measured genotype data were phased using SHAPEIT[52] and imputed to the 1000GP data using Minimac3[53]. SNPs with MAF $>1\%$ were retained, leaving a total of 9,377,008 SNPs.

**Peripheral blood samples.** Samples were obtained from the Study to Explore Early Development (SEED), a multi-site, national case-control study of children aged 3–5 years with and without ASD. Overall, 2800 families were recruited and classified into three groups according to the status of the child: the ASD group, the general population control group, and the (non-ASD) developmental delay group[54]. This study was approved as an exemption from the Johns Hopkins IRB under approval 00000011. Informed consent was obtained from all participants as part of the parent SEED study. SEED recruitment was approved by the IRBs of each recruitment site: IRB-C, CDC Human Research Protection Office; Kaiser Foundation Research Institute (KFRI) Kaiser Permanente Northern California IRB, Colorado Multiple IRB, Emory University IRB, Georgia Department of Public Health IRB, Maryland Department of Health and Mental Hygiene IRB, Johns Hopkins Bloomberg School of Public Health Review Board, University of North Carolina IRB and Office of Human Research Ethics, IRB of The Children's Hospital of Philadelphia, and IRB of the University of Pennsylvania. All enrolled families provided written consent for participation.

**Peripheral blood DNA methylation.** Genomic DNA was isolated from whole blood samples using the QIAsumphonia midi kit (Qiagen, Valencia, CA). For each a subset of case and control samples ($n = 630$), bisulfite treatment was performed using the 96-well EZ DNA methylation kit (Zymo Research, Irvine, CA). Samples were randomized within and across plates to minimize batch and position effects. The minfi R package (version 1.16.1) was used to process Illumina.idat files generated from the array[45]. Control samples ($n = 14$) were removed and quantile normalization performed using the minfi function preprocessQuantile()[55]. Probes with failed detection $P$ value ($> 0.01$) in $>10\%$ of samples were removed ($n = 772$), as were probes annotated as cross-reactive ($n = 29,233$)[48], and probes on sex chromosomes ($n = 11,648$). Samples were excluded if reported sex did not match

predicted sex (minfi function getSex()) ($n = 0$), detection $P$ values > 0.01 in >1% of sites ($n = 2$), low overall intensity (median methylated or unmethylated intensity <11; $n = 2$), and if they were duplicates ($n = 8$). Successive filtering according to these criteria resulted in 445,154 probes and 604 samples.

**Peripheral blood genotyping.** Of the SEED samples with DNAm data, 590 had whole-genome genotyping data available, measured using the Illumina HumanOmni1-Quad BeadChip (Illumina, San Diego, CA). Standard quality control measures were applied: removing samples with < 95% SNP call rate, sex discrepancies, relatedness (Pi-hat > 0.2), or excess hetero- or homozygosity; removing markers with <98.5% call rate, or monomorphic. Phasing was performed using SHAPEIT[52] followed by SNP imputation via the IMPUTE2 software[53], with all individuals in the 1000 Genomes Project as a reference. Genetic ancestry was determined using EigenStrat program[56]. A total of 4,948,723 SNPs were available post imputation at MAF > 1%.

**Normal lung tissue meQTLs.** A list of meQTLs identified in a recent characterization of normal lung tissue[23] as well as the total list of SNPs ($n = 569,753$) and 450k CpG sites ($n = 338,730$) tested for meQTL identification (i.e., passed filtering and QC done in that study) was obtained from the study authors.

**Fetal brain meQTLs.** Fetal brain meQTLs were identified via imputed genotypes in a recent study examining meQTLs in the context of schizophrenia[22]. The total list of SNPs ($n = 5,159,699$) and 450k CpG sites ($n = 314,554$) that were tested (i.e., passed filtering and QC done in that study) was obtained from the study authors. For all analyses, only fetal brain meQTLs within a SNP to CpG distance of 1 Mb, were included, in order to improve comparability to the other three meQTL lists, where distant (trans) meQTL relationships were not explored (peripheral blood, cord blood) or used (lung).

**meQTL identification parameters.** There are three main parameters of interest in a meQTL query: the SNP MAF threshold for inclusion, the definition of standard deviation cutoff that dictates a CpG site is variably methylated, and the maximum physical distance between a SNP and CpG site to be queried, often referred to as the window size. These 3 factors contribute to the total number of CpG to SNP linear regression tests that are performed. Our available sample sizes (and thus statistical power, at fixed effect size) for the joint DNAm and genotype data differed for peripheral blood ($n = 339$) and cord blood ($n = 121$) analyses after limiting both to samples of European descent identified via principle components analysis of the SNP data. Thus, the ideal combination of these parameters should differ between the two study populations for them to be comparable.

For each tissue sample set, we computed the total number of CpG-to-SNP linear regression tests at various combinations of the three main parameters of interest to a meQTL query. We then used the genetic power calculator Quanto[57] to determine the most permissive set of parameters that allowed for 80% power to detect a 5% difference in methylation for each addition of the minor allele, at the lowest allowed MAF. We computed this power calculation at a Bonferroni-based significance level derived from the total number of CpG to SNP linear regression tests. We defined 'most permissive' in a hierarchical manner that first prioritized the inclusion of the most methylation sites (lowest sd cutoff), then the inclusion of the most number of SNPs (lowest MAF threshold), and then the use of the largest window size. This procedure resulted in study-specific MAF thresholds for the SNP data, standard deviation cutoffs for the methylation data, and window sizes that were tailored to the number of samples available.

**meQTL identification procedure.** Pairwise associations between each SNP and CpG site were estimated via the R package MatrixEQTL[58], with percent methylation (termed 'Beta value', ranging from 0 to 100) regressed onto genotype assuming an additive model, adjusting for the first two principal components of ancestry and sex. Models did not adjust for age given the very narrow age ranges in each tissue type.

FDR was controlled via permutation[23]. Briefly, the total number of CpG sites ($N_{obs}$) under genetic control was obtained for a meQTL $P$ value of $p_o$. Genome-wide meQTL query was performed for each of 100 permuted sets of the genotype data (scrambling sample IDs, to retain genotype correlation structure). In each set, we retained the total number of CpG sites under genetic control ($N_{null}$) at the same $P$ value $p_o$. The FDR was defined as the mean ($N_{null}$)/$N_{obs}$. Finally we determined the value of $p_o$ to control the FDR at values of 10, 5, and 1%. Both the meQTL discovery and FDR determination were performed in each tissue or study sample.

**Enrichment of meQTLs in ASD-associated SNPs.** We tested for enrichment of meQTLs from four tissue types among ASD GWAS SNPs. ASD SNPs were assigned from the PGC-AUT analysis (downloaded February 2016), based on 5305 cases and 5305 pseudocontrols[59]. The PGC provides results for 9,499,589 SNPs; 11,749 SNPs exceeded an ASD $P$ value threshold of 1e–03, and 1094 SNPs exceeded an ASD $P$ value threshold of 1e–04. For each tissue, we included only SNPs available in both the PGC-AUT analysis and our meQTL analysis, either via

direct or proxy ($r^2 > 0.8$ within 500 Kb window in CEU 1KG) overlap as defined via the SNAP software[60].

To estimate the proportion of meQTLs among ASD SNPs vs. among all SNPs (or a sample of null SNPs), we recognized three important factors that could differ between null SNP sets and the ASD SNP set: LD structure, MAF distribution, and number of CpG sites per window size. We designed a comparison process to address each of these. First, we performed LD pruning 'supervised' by PGC ASD $P$ value (so as to not prune away all ASD SNPs) using PriorityPruner (v0.1.2)[61], removing SNPs at $r^2 > 0.7$ within a sliding 500 Kb window. For the peripheral blood and cord blood data sets this pruning was done with the study-specific genotype data, and for the fetal brain and lung data sets this pruning was done with 1000 Genomes CEU samples. Second, we grouped remaining SNPs into MAF bins of 5%. Third, we characterized each SNP according to the number of CpGs within the meQTL discovery window size to allow for differential opportunity to have been identified as a meQTL. We then collapsed this number into categories of 0–49, 50–99, etc. to reflect the same concept. We defined 1000 null SNP sets by finding, for each SNP in the ASD set, a random SNP in the genome that matched that SNP on both MAF bin and CpG opportunity. We computed an enrichment fold statistic as the proportion of meQTLs in the ASD SNP set divided by the mean proportion of meQTLs across null sets; and a $P$ value as the total number of null set proportions as or more extreme than in the ASD set. To evaluate the robustness of our results, we used two PGC AUT $P$ value cutoffs (1e–03, 1e–04) and three meQTL $P$ value cutoffs (FDR 10%, 5%, 1%) for peripheral blood and cord blood. However, based on available information for lung and fetal brain, we were limited to assess our results at FDR 5% for lung, and $P < 1e–08$ for fetal brain for meQTL $P$ value.

**Gene ontology analysis of meQTL targets.** We further assessed which biological pathways are implicated by the location of CpG targets of ASD SNPs acting as meQTLs. To do this, we identified CpG sites associated with ASD SNPs (ASD-related meQTL targets) among all CpG sites controlled by SNPs (all meQTL targets). We then examined Gene Ontology (GO) terms specific to these ASD-related meQTL targets, in order to enumerate biological pathways engaged specifically by ASD SNPs.

Specifically, we first filtered the full list of CpG sites associated with any meQTL to only those sites associated with an ASD SNP (PGC Wald $P < 1e–04$; $N = 1094$) or their proxies ($r^2 > 0.8$ within 500 kb window in CEU 1KG as defined via the SNAP software[60]). We used thresholds of FDR $\leq$ 5% for peripheral and cord blood meQTL lists, and Wald $P < 1e–08$ for the fetal brain list. We only examined CpG sites that did not overlap with SNPs within 10 bp of the CpG site or at the single base extension[62], as it has been previously demonstrated that these CpG sites may strongly influence functional-type enrichment analysis of CpG sites[63], and these CpG sites were not examined in the fetal brain meQTL lists[22]. We used the gometh () function in the MissMethyl R package[64], which maps 450k DNAm sites to their nearest gene, and corrects for bias due to non-uniform coverage of genes on the 450k. We further ran nominally significant (hypergeometric test $P < 0.05$) results for the category "biological processes" through the REVIGO tool to avoid reporting GO terms with a greater than 70% overlap in gene lists[65]. Finally, we determined the set of terms in these lists that overlapped at least two tissues, and prioritized them by summing the scaled, enrichment $P$ value-based rank in each tissue. This scaling was done by dividing the raw rank for the term in the list for that tissue by the total number of nominally significant, post-REVIGO terms for that tissue.

We also ran analogous GO analyses comparing all meQTL targets to all CpGs to explore functional implications for meQTL targets vs. CpGs not under strong genetic control. This allowed for comparison of ASD SNP-specific functional pathways engaged through methylation vs. general SNP functional pathways engaged through methylation.

**Identifying genes via ASD meQTL target locations.** Defining ASD SNPs as those with PGC Wald $P < 1e–04$, meQTL relationships as in the GO analysis, and RefSeq genes from the UCSC Genome Browser[66], we annotated gene overlap (if any) via findOverlaps() in the GenomicRanges R package[66] for all ASD SNPs and their associated CpG sites (if any). We filtered out long intergenic non-coding RNAs, long non-coding RNAs, microRNAs, and small associated RNAs from the RefSeq gene list. We further collapsed SNPs into bins by LD block. Blocks were defined using recombination hot spot data from 1000 Genomes[51].

**Regulatory feature characterization of meQTL targets.** To quantify the propensity of regulatory features to overlap with meQTL targets within and across tissue type, we first compared regulatory feature overlap of all meQTL targets to non-meQTL targets. We next compared meQTL targets of psychiatric condition-related SNPs to meQTL targets of SNPs unrelated to psychiatric conditions. SNPs associated with psychiatric conditions were obtained from the PGC cross-disorder analysis[17] (PGC Wald $P < 1e–04$) and their proxies. We used these SNPs in order to analyze a greater total number of meQTL targets than associated with ASD SNPs only (and thus ensure a well-powered analysis), and to make functional insights that could be applied to psychiatric disease more broadly. Our results are still relevant to ASD in light of known cross-disorder GWAS consistency[16].

We performed both comparisons for unique and overlapping tissue categories ($n = 7$): peripheral blood, cord blood, fetal brain, lung, intersection of peripheral blood and cord blood, intersection of peripheral blood and fetal brain, and intersection of peripheral blood and lung. For each intersection, we conducted a meQTL discovery screen in which the peripheral blood was down sampled to the sample size of the other tissue, and run at the same parameters used to identify meQTLs in that tissue. This increases comparability with respect to power and meQTL query parameters. For the peripheral blood overlaps with cord blood and lung, we also computed the meQTL $P$ value to control the FDR at 5%, since this FDR threshold was available for each of those tissue types[23]. However, we only computed the FDR $P$ values using the data from the first 6 chromosomes, as we found empirically that FDR $P$ value estimates stabilized by this point. Finally, for the peripheral blood-fetal brain comparison, we retained results for peripheral blood that passed a meQTL $P$ value of 1E-8, as reported from the fetal brain study[22].

Regulatory feature information came from several sources. General DHSs were defined as those CpG probes experimentally determined to be within a DHS, as determined by the manifest for the 450k array[67]. In addition, the tissue-specific DHS data were tested for enrichment. Brain DHSs were downloaded from GEO[68] for three brain regions: Frontal Cortex [GEO Sample ID: GSM1008566], Cerebellum [GSM1008583], and Cerebrum [GSM1008578]). Two blood (CD14+ Monocytes; 'wgEncodeOpenChromDnaseMonocd14' and CD4+ cells; 'wgEncodeUwDnaseCd4naivewb78495824PkRep1') and one lung-derived (IMR90; 'wgEncodeOpenChromDnaseImr90Pk') data sets were additionally downloaded from the UCSC Genome Browser[66].

The tissue-specific histone data were compiled from the Roadmap Epigenomics Project[69] for five different marks: H3K27me3, H3K36me3, H3K4me1, H3K4me3, and H3K9me3. As the Epigenome Roadmap Project data were often generated across a number of individuals, for those cases in which data were generated in more than one Caucasian individual, the overlap across individual samples was utilized in downstream analyses. Overlap was calculated using the UCSC Genome Browser's 'intersect' function for those samples indicated in Table 4. Regions with any overlap were included in functional enrichment analyses.

Finally, TFBS information from ChIP-Seq experiments carried out by the ENCODE project[70] were extracted for 161 transcription factors from the UCSC Genome Browser ('wgEncodeRegTfbsClusteredV3')[66].

Significant feature overlap was assessed via two-sided Fisher's $2 \times 2$ exact test, with Bonferroni correction ($P < 0.05/(181$ regulatory features $\times 7$ categories$)$) = 3.95e−05). Odds ratio and $P$ value were recorded for each test in each unique and overlapping tissue category.

**Code availability**. Scripts for the analyses conducted in this study are available at https://github.com/sandrews5/ASDmeQTL_Manuscript.git

**Data availability**. The data from the EARLI study is available from the National Database for Autism Research repository (collection number: 2462). SEED data are not available at this time due to restrictions imposed by the informed consent signed by the study participants. We are working with the Centers for Disease Control and Prevention (CDC) to find a solution that might enable deposition of these data into a genomics data repository in the future. All other data that support the findings of this study are available from the corresponding authors upon reasonable request.

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

## Acknowledgements

We thank Eilis Hannon and Jonathan Mill for sharing the complete list of SNPs and 450k probes tested in their fetal brain meQTL analysis. We thank Jianxin Shi for sharing the same list from his lung meQTL study. S.V.A. was supported by the Burroughs-Wellcome Trust training grant: Maryland, Genetics, Epidemiology and Medicine (MD-GEM). The EARLI study was supported by NIEHS R01ES016443 and Autism Speaks grant #260377. The SEED study was supported in part by Autism Speaks #7659, NIEHS (R01ES019001; R01ES017646), and the Centers for Disease Control and Prevention (U10DD000180, U10DD000181, U10DD000182, U10DD000183, U10DD000184, U10DD000498).

## Author contributions

C.L.-A. and M.D.F.: Conceived the study. M.D.F., C.J.N., L.A.C., and I.H.-P.: Led participation recruitment and sample selection for the EARLI study. S.V.A.: Performed quality control and processing for peripheral blood methylation data. B.S. and C.L.-A.: Performed quality control for the peripheral blood genotype data. K.M.B.: Performed quality control and processing for the cord blood methylation and cord blood genotype data. S.V.A.: Designed meQTL parameter selection process, performed the meQTL queries, and implemented the FDR estimation. S.V.A. and S.E.E.: Designed and conducted all SNP and CpG-based enrichment analyses. S.V.A.: Designed and conducted GWAS loci expansion analysis. D.E.A, C.L.-A, and M.D.F.: Supervised all analyses. S.V.A, S.E.E., K.M.B., C.L.-A, and M.D.F.: Contributed to writing the manuscript. All authors contributed to interpretation of results and edited and reviewed the manuscript.

## Additional information

Competing interests: The authors declare no competing financial interests.

