## [Peer Review File · Nature Communications]

Reviewers' comments:

Reviewer #2 (Remarks to the Author):

The authors of the Andrews et al manuscript analyzed the overlap between SNPs associated with autism in ~5000 case and ~5000 control association study (autSNPs) with SNP associated with levels of methylation of nearby CpG sites identified independently through few hundred samples in each of 4 different tissues: cord blood, peripheral blood, fetal brain tissue, and lung tissue (meQTLs). The genotype and methylation data for the cord and the peripheral bloods are generated through hybridization to the appropriate chip designs as part of this effort while the fetal brain and lung meQTLs were taken from previously published manuscripts from independent groups. To increase the power to detect overlaps the method uses imputation and extends the associated SNPs with strongly correlated and linked proxy SNP. The significance of the overlap is measured by a permutation method that controls for several properties of the observed autSNPs and meQTLs like minor allele frequency, number of nearby CpG sites, etc. The authors also test the overlaps under several cutoff for identifying autSNPs and meQTLs.

Healthy, statistically significant, overlap is observed between autSNPs and fetal brain and peripheral blood meQTLs, weak overlap is observed between autSNPs and cord blood meQTL, and no significant overlap is observed between autSNPs and lung meQTLs.

The authors implicitly assume that the effect on autism of the autSNPs&meQTL SNPs (the SNPs that are both associated with autism and with the methylation of one or more CpG sites) is conferred through the methylation of status of the associated CpG sites and follow up the significant-overlap observation with GO analysis of the genes harboring such CpG sites and with 'regulatory features' associated with such CpG sites. My largest concern with the manuscript is that this main assumption is not made explicitly. It would improve the manuscript if the authors justify it by the prior observations or with the results reported here.

The core of the paper, that includes identification of meQTLs for the two blood samples and tests for overlap between autSNPs and meQTLs seem technically sound. I'm less impressed with the GO annotation test that is too complicated for me to understand and evaluate. But, I have bigger issues with the test for 'regulatory features'. The switch from autism to all neuropsychiatric disorder seems strange and I imagine that it was done because there were no positive results for the autism SNPs. It would be inappropriate if that were the case. Also, the table with results of the 'regulatory properties' of the all meQTL CpG sites vs the non-meQTL CpG sites shows that almost all of the regulatory features are highly significant. I find that surprising and it might point to a bug. If it's not due to a bug, isn't it a very important observation that needs to be prominently presented in the manuscript?

Minor:

We used publicly available lung (to include a likely non ASD-related tissue) and fetal brain meQTL lists and ??thus?? the p-value cutoffs stated in those respective studies ($P = 1e-08$ for fetal brain and $P = 4e-05 = \text{FDR } 5\%$ for lung)

threshold (enrichment fold = 1.70, Penrichment < $1e-03$, PGWAS < $1e-03$), and at a more ??stringent?? GWAS

Reviewer #4 (Remarks to the Author):

The authors present an interesting approach to integrating epigenomic and genomic information to

further understand previously implicated GWAS hits for autism spectrum disorder while at the same time addressing concerns about tissue relevance and specificity. They argue that looking for meQTLs related to GWAS SNPs provided further information about additional genetic candidates as well as implicating potential biological functions.

The paper as currently written is quite dense. Generally the authors present the reader with a lot of options for evaluating their data. While nice, at the end of the day, it would be better if the authors presented clearly their best interpretation of the results (example would be sticking to one FDR for interpretation of total number of meQTLs that they believe are meaningful.)

Specific comments:

Line 181: what are the implications of this statement? This was not discussed in the discussion section and yet would seem to imply that it is not specific to brain related disease?

Line 395; the authors refer to study specific thresholds. Is it possible, for instance, that some results were only possible in one tissue type based on these differing cutoffs?

Line 414: what is the universe of ASD SNPs used to begin with? Please include the number somewhere.

Line 418; the authors state that enrichment was calculated based off of all SNPs in the denominator or null SNPs. Later they discuss identifying sets of 1000 null SNPs. So it would seem they never used 'all' SNPs. Please clarify.

Line 435: The authors present 3 FDR cutoff levels. It is unclear which the authors ultimately choose to present. It would be helpful to the reader to state it clearly here.

Table 2. why present 3 different FDRs when the values are virtually the same across them? Why not stick with FDR 5% to match the Lung data for which that is all they have.

Figure 1: I recognize the desire to visually summarize or represent the meQTLs but it is impossible for this reviewer to make sense of this figure in any meaningful way.

IN figure 2; can the authors comment on what the implications are of the fact that the ORs in the black box are generally lower than the ORs in the gray boxes?

Wording in footnotes of some tables seems awkward or missing words

Reviewer #5 (Remarks to the Author):

Summary: This is an interesting study that integrates genome-wide methylation and genetic data with pre-existing GWAS data on ASD to identify novel genes using quantitative trait association analyses to relate the degree of methylation at discrete CpG targets with ASD-associated SNPs identified by the Cross-disorder group of the Psychiatric Genomics Consortium. The use of several tissue types for these analyses reveals both common and tissue-specific associations, with the fetal brain and peripheral blood providing the greatest enrichment of meQTLs associated with ASD-SNPs. While describing an impressive set of analyses, the manuscript could be improved with the following additions; in particular, more information on the specific genes implicated by the meQTLs would allow a better sense of the relationship of these "meQTL targets" to ASD.

Major points to be addressed:

1. Table 1: Include an additional column showing the # of discrete genes associated with the meQTL targets.
2. Table 2: Are the p-values for the ASD-associated SNPs from the PGC study nominal or corrected for multiple testing? If nominal, can the authors justify the use of relatively liberal p-values (i.e., $1e-03$ and $1e-04$)?
3. With respect to the gene ontology analyses presented in Table 3, it would be much more informative to also present the lists of genes used for the GO analyses for each tissue.

4. Related to point 3, how do the genes implicated by the ASD-SNP associated meQTLs in this study compare with those identified by other methylation studies of ASD?
5. Fig. 1: What is/are the relationship(s) between the genes shown in Figure 1 and genes already associated with ASD?
6. Discuss the significance/meaning of the greatest number of ASD-SNP associated meQTLs in peripheral blood relative to cord blood and fetal brain.
7. Fig. 2: The enrichment of meQTL targets in DNase hypersensitivity sites shown for PGC meQTL targets vs. non-PGC meQTL targets is very interesting. What is the analogous DHS enrichment data for ASD-meQTL targets vs. the non-PCG meQTL targets?

Minor points:

8. Define what is meant by "meQTL targets"
9. Page 14, 1st paragraph, line 6: "The 232 mothers with a subsequent sibling..." should be "...subsequent child..."
10. What were the total numbers of fetal brain and lung tissues analyzed for meQTLs?
11. Page 18, second paragraph, last sentence: "...to be comparable." should be "...to be compared."
12. Page 23, last paragraph, line 4: "...were downloaded in from GEO..." Remove "in".
13. Footnote to Table 3: "NA shown for terms that did appear..." I think the authors meant "...did not appear..."

Reviewer #2 (Remarks to the Author):

The authors implicitly assume that the effect on autism of the autSNPs&meQTL SNPs (the SNPs that are both associated with autism and with the methylation of one or more CpG sites) is conferred through the methylation of status of the associated CpG sites and follow up the significant-overlap observation with GO analysis of the genes harboring such CpG sites and with 'regulatory features' associated with such CpG sites. My largest concern with the manuscript is that this main assumption is not made explicitly. It would improve the manuscript if the authors justify it by the prior observations or with the results reported here.

RESPONSE:

We agree that these assumptions should be explicit. We added the following underlined phrases in the introduction:

“Previous studies of bipolar disorder, schizophrenia, and ASD have demonstrated the enrichment of GWAS results for expression quantitative trait loci (eQTLs), suggesting functional biology of GWAS SNPs, assuming those SNPs confer some risk through regulatory mechanisms. Given the implications of epigenetic regulation in ASD from rare variant findings^{6,8}, the epigenetic basis of ASD-related conditions, and the association of histone modifications and DNA methylation in multiple tissues, similar examination of epigenetic marks is an important next step towards prioritization and characterization of ASD genetic results, assuming similarly that ASD genetic risk may act in part through epigenetic regulation. As with expression loci, genetic variation contributes to DNAm levels locally and distally^{22,23} and thus integration of methylation quantitative trait loci (meQTLs), or SNPs that are highly associated with DNAm, and autism-associated GWAS results may inform our understanding of autism GWAS findings. Moreover, meQTLs are enriched in top hits for bipolar disorder¹⁴ and schizophrenia^{15,22}, which has a well-established genetic overlap with ASD¹¹.”

The core of the paper, that includes identification of meQTLs for the two blood samples and tests for overlap between autSNPs and meQTLs seem technically sound. I'm less impressed with the GO annotation test that is too complicated for me to understand and evaluate.

RESPONSE:

We have tried to simplify our explanation of the GO analytic approach with an introductory paragraph in the relevant methods section:

“We further assessed which biological pathways are implicated by the location of CpG targets of ASD SNPs acting as meQTLs. To do this, we identified CpG sites associated with ASD SNPs (ASD-related meQTL targets) among all CpG sites controlled by SNPs (all meQTL targets). We then examined Gene Ontology (GO) terms specific to these ASD-related meQTL targets, in order to enumerate biological pathways engaged specifically by ASD SNPs. “

But, I have bigger issues with the test for 'regulatory features'. The switch from autism to all neuropsychiatric disorder seems strange and I imagine that it was done because there were no positive results for the autism SNPs. It would be inappropriate if that were the case.

RESPONSE:

We did not do an ASD-specific meQTL target enrichment analysis for this. Our primary goal in this specific features analysis was to characterize *cross-tissue* meQTL targets with respect to their functional propensity, because we could not find such information yet in the literature. Doing this analysis in the context of a broader psychiatric phenotype allowed for a greater number of GWAS SNPs to be used and therefore a greater number of relevant meQTL targets. This in turn ensured an adequate number of meQTL targets that overlapped across tissues to enable a meaningful enrichment analysis. Therefore, while we hope to ask these same regulatory feature enrichment questions in the context of ASD specifically, current data identifying ASD meQTL targets precludes a well-powered analysis, so we did not attempt it. Our results are still relevant to ASD in light of known cross-disorder GWAS consistency¹. We have added the following underlined phrases to this passage from the Methods section ‘Regulatory feature characterization of meQTL targets’:

“We used these SNPs in order to analyze a greater total number of meQTL targets than associated with ASD SNPs only (and thus ensure a well-powered analysis), and to make functional insights that could be applied to psychiatric disease more broadly. Our results are still relevant to ASD in light of known cross-disorder GWAS consistency¹¹.”

1. *Cross-Disorder Group of the Psychiatric Genomics Consortium et al. Genetic relationship between five psychiatric disorders estimated from genome-wide SNPs. Nat. Genet. 45, 984–994 (2013).*

Also, the table with results of the ‘regulatory properties’ of the all meQTL CpG sites vs the non-meQTL CpG sites shows that almost all of the regulatory features are highly significant. I find that surprising and it might point to a bug. If it’s not due to a bug, isn’t it a very important observation that needs to be prominently presented in the manuscript?

RESPONSE:

We pursued this analysis to compare within and across tissue psychiatric meQTL target enrichment results with what would be seen for meQTL target generally. An important feature of this all meQTL-targets analysis is the massively increased sample size – 1-2 orders of magnitude greater than when restricting to psychiatric meQTLs. Thus, nearly any non-zero enrichment effect will be statistically significant. If one instead focuses on the effect size estimates, realizing that the significance somewhat loses meaning in this large N situation, we see that the within tissue meQTL target functional enrichment has very small effects. It is only the cross-tissue enrichment results that show more moderate effects. We believe this is consistent with the idea that genetically controlled CpGs are more likely to have common function across tissues (given they all share genomic sequence) compared those less under direct genetic control. For example, one study has demonstrated that cross-tissue meQTLs (the SNPs, rather than meQTL CpG targets) are enriched for miRNA binding sites². There is not literature, that we can find, that has examined this for meQTL targets themselves. We did not highlight this general meQTL target function finding in the current paper, since the main focus was to use these results as a comparison for our psychiatric meQTL functional enrichment, and it seemed too distracting to go down this tangent. However, as pointed out by the reviewer and

our note of the lack of literature on this point, we do think it is important to present these full results as supplementary data to benefit researchers who are interested.

We did examine our pipeline thoroughly at multiple steps and do not believe this is the result of a coding bug.

We have modified the results section:

“When considering meQTL targets generally (regardless of their status of being downstream of PGC SNPs), compared to non-meQTL-target CpGs, enrichment was observed for DHSs for all 7 meQTL target lists, with the largest effect sizes among the peripheral blood-cord blood overlap list and the peripheral blood-fetal brain overlap list (gray font and boxes, Fig. 2). In fact, a large number of regulatory features were statistically significantly enriched among meQTL targets for each tissue, including lung, given the very large sample size (Supplementary Data 8). However, these were typically small effects. Larger, but still moderate enrichment effects were primarily seen for cross-tissue overlap meQTL lists, particularly for the peripheral blood-cord blood overlap list and the peripheral blood-fetal brain overlap list. “

We have added to the discussion:

“We also considered regulatory feature overlap with meQTL targets in general, for comparison to results for psychiatric meQTLs. Given the very large number of meQTL targets when not restricting to those downstream of psychiatric GWAS SNPs, many cell type specific DHS sites and chromatin marks showed significant enrichment for meQTLs in general. Among within-tissue analyses, the effect sizes were small. However, enrichment seen in meQTL targets that overlapped peripheral blood and cord blood, and meQTL targets that overlapped peripheral blood and fetal brain was of larger effect size (though still moderate). This is consistent with the idea that genetically controlled CpGs are more likely to have common function across tissues (given they all share genomic sequence) compared to those less under direct genetic control. For example, one study has demonstrated that cross-tissue meQTLs (the SNPs, rather than meQTL CpG targets) are enriched for miRNA binding sites². We have not found literature examining cross-tissue meQTL targets themselves. This finding suggests an important area for future research, that could give greater context to our and future investigations of cross-tissue meQTL targets in a disease context specifically.”

2. Smith, A. K. et al. Methylation quantitative trait loci (meQTLs) are consistently detected across ancestry, developmental stage, and tissue type. *BMC Genomics* **15**, 145 (2014).

Minor:

We used publicly available lung (to include a likely non ASD-related tissue) and fetal brain meQTL lists and thus the p-value cutoffs stated in those respective studies ($P = 1e-08$ for fetal brain and $P = 4e-05 = \text{FDR } 5\%$ for lung) threshold (enrichment fold = 1.70, Penrichment < $1e-03$, PGWAS < $1e-03$), and at a more stringent GWAS

RESPONSE:

We thank the reviewer for pointing out these errors. We have corrected them in the revised manuscript.

Reviewer #4 (Remarks to the Author):

The paper as currently written is quite dense. Generally the authors present the reader with a lot of options for evaluating their data. While nice, at the end of the day, it would be better if the authors presented clearly their best interpretation of the results (example would be sticking to one FDR for interpretation of total number of meQTLs that they believe are meaningful.)

RESPONSE:

We tried to pare down results provided, with prioritized results in the paper body and further results in supplemental material. As consumers of similar scientific literature, we have greatly appreciated broad details and results provided to readers for re-analysis and evaluation, so have left material in our work, with that in mind. We are striving to make this work as reproducible as possible by including all relevant details.

We do empathize with the need to state our best interpretation of results, and have attempted to do this as honestly as possible, but feel that multiple scenario results and interpretations are necessary in some cases. For example, we report results for multiple thresholds in the SNP-based enrichment analysis because validity of these results is directly tied to their robustness across ASD and meQTL p-value thresholds. Interpretation of peripheral blood and fetal brain results are strengthened by their consistency across thresholds. Had we selected a single set of cutoffs, we may have come to a different conclusion. Further, previous meQTL/eQTL enrichment analyses for bipolar disorder³ and schizophrenia^{4,5} also used multiple thresholds in this manner, and we designed our study to be complementary in the context of ASD. When reasonable, we did select a single threshold, such as the meQTL p-value (FDR = 5%) and ASD p-value (1E-4) thresholds for the meQTL target enrichment and GWAS loci expansion analyses.

We have added the following line to the discussion:

“When considering the ASD SNPs, we found enrichment of fetal brain and peripheral blood meQTLs which strengthened in magnitude at increasingly stringent meQTL p-value and ASD p-value thresholds, bolstering confidence in these findings.”

3. Gamazon, E. R. et al. *Enrichment of cis-regulatory gene expression SNPs and methylation quantitative trait loci among bipolar disorder susceptibility variants. Mol. Psychiatry* **18**, 340–346 (2013).

4. Hannon, E. et al. *Methylation QTLs in the developing brain and their enrichment in schizophrenia risk loci. Nat. Neurosci.* **19**, 48–54 (2016).

5. van Eijk, K. R. et al. *Identification of schizophrenia-associated loci by combining DNA methylation and gene expression data from whole blood. Eur. J. Hum. Genet. EJHG* **23**, 1106–1110 (2015).

Specific comments:

Line 181: what are the implications of this statement? This was not discussed in the discussion section and yet would seem to imply that it is not specific to brain related disease?

RESPONSE:

Please see response to reviewer 2, regarding regulatory feature enrichment, above. We have modified results and discussion in the manuscript to make the purpose of these analyses and their interpretation clearer.

Line 395; the authors refer to study specific thresholds. Is it possible, for instance, that some results were only possible in one tissue type based on these differing cutoffs?

RESPONSE:

The comparability across tissue types was in fact the motivation for our data-driven approach because arbitrary thresholds that are fixed across different study samples (i.e. tissue types here) are not comparable due to differences in sample sizes, detectable effect sizes, and QC pipeline differences. This approach, seeking to harmonize detection “power” across samples, necessarily leads to different thresholds across the key parameters per sample. This could indeed affect some results - for example, a minor allele frequency threshold of 5% would preclude detection of alleles with MAF < 5%. We were comfortable with this trade off, choosing overall comparability of meQTL lists versus specific scenarios.

Line 414: what is the universe of ASD SNPS used to begin with? Please include the number somewhere.

RESPONSE:

We thank the reviewer for mentioning this point. We have added the following sentence to the Methods section, in the ‘Enrichment of meQTLs in ASD-associated SNPs’:

“The PGC provides results for 9,499,589 SNPs; 11,749 SNPs exceeded an ASD p-value threshold of 1e-03, and 1,094 SNPs exceeded an ASD p-value threshold of 1e-04.”

Line 418; the authors state that enrichment was calculated based off of all SNPs in the denominator or null SNPs. Later they discuss identifying sets of 1000 null SNPs. So it would seem they never used ‘all’ SNPs. Please clarify.

RESPONSE:

We used sets of 1000 randomly drawn null SNPs to represent the concept of “all SNPs”. We hope this is clear in the methods.

Line 435: The authors present 3 FDR cutoff levels. It is unclear which the authors ultimately choose to present. It would be helpful to the reader to state it clearly here. Table 2. why present 3 different FDRs when the values are virtually the same across them? Why not stick with FDR 5% to match the Lung data for which that is all they have.

RESPONSE:

Please see previous comment discussing our rationale for presenting results at multiple FDR thresholds.

Figure 1: I recognize the desire to visually summarize or represent the meQTLs but it is impossible for this reviewer to make sense of this figure in any meaningful way.

RESPONSE:

We borrowed this presentation concept from multiple other published papers by our group and colleagues^{4,6,7}. We respectfully argue that it does show the key points of our results, namely that meQTL targets can be used to effectively ‘expand’ GWAS-identified loci beyond locations of the SNPs alone to also include positions of the meQTL target CpG sites.

However, we empathize with a peer who does not find this useful in its current form. We have removed the CpG-CpG correlation portion of the panels – since these are not specifically mentioned in results – allowing better focus on the location of meQTL targets with respect to annotated genes, as this is the main utility of the figure. We have also revised language in the paper to emphasize this ‘expansion’ point wherever possible, and revised the figure title and legend for additional clarity:

“Figure 1: ‘Expansion’ of ASD-related PGC loci through meQTL target mapping in peripheral blood, cord blood, and fetal brain. Each tissue-specific panel presents, from bottom to top: genomic location, gene annotations, SNP locations, SNP-CpG associations, CpG locations. Light gray meQTL association lines denote all SNP to CpG associations in that tissue type; Dark meQTL association lines denote SNP-CpG associations for ASD-associated SNPs in PGC (p -value $\leq 1e-04$). Panel A) Locus at chr8; Panel B) Locus at chr19. Data are presented for meQTL maps for fetal brain (top); cord blood meQTLs (middle), and peripheral blood meQTLs (bottom). Please note locus coordinates differ from those in Supplementary Data 6 because in this context they encompass the locations of meQTL target CpG sites.

We hope that these revisions will help guide readers through the visualization with more ease.

4. Hannon, E. et al. Methylation QTLs in the developing brain and their enrichment in schizophrenia risk loci. *Nat. Neurosci.* **19**, 48–54 (2016).

6. Liu, Y. et al. Epigenome-wide association data implicate DNA methylation as an intermediary of genetic risk in rheumatoid arthritis. *Nat. Biotechnol.* **31**, 142–147 (2013).

7. Liu, Y. et al. GeMes, clusters of DNA methylation under genetic control, can inform genetic and epigenetic analysis of disease. *Am. J. Hum. Genet.* **94**, 485–495 (2014).

IN figure 2; can the authors comment on what the implications are of the fact that the ORs in the black box are generally lower than the ORs in the gray boxes?

RESPONSE:

We would hesitate to over-interpret observed differences between the gray and black boxes. We did not explicitly test for differences between these ORs because the precision on estimates in the gray boxes is so much greater than for the black boxes (there is a much greater number of meQTL targets than those that are downstream of PGC-psych SNPs specifically). We provide the gray box results (based on all meQTL targets) as context for interpretation of the psychiatric SNP-based meQTL targets in the black boxes, but did not intend to make inference about any observed differences in value.

Wording in footnotes of some tables seems awkward or missing words

RESPONSE:

We thank the reviewing for pointing this out. We have revised the footnotes of all tables.

Reviewer #5 (Remarks to the Author):

While describing an impressive set of analyses,

RESPONSE:

We are happy to see the reviewer's appreciation for the broad set of analyses.

the manuscript could be improved with the following additions; in particular, more information on the specific genes implicated by the meQTLs would allow a better sense of the relationship of these "meQTL targets" to ASD.

RESPONSE:

We debated how much information on specific genes to include in this manuscript. We do provide locations and gene names for all results in supplementary data. Our hesitation regarding dedicating text to specific genes in the manuscript was based on the interim nature of our current PGC GWAS ASD list. Specific ASD SNP meQTL targets are likely to be further refined once our approach can be applied to an updated ASD GWAS list. In the meantime, our enrichment results are themselves the key point of this paper, rather than any specific gene target. We retained the region-specific results to inform the reader of the concepts of our method and as proof of principle.

Major points to be addressed:

1. Table 1: Include an additional column showing the # of discrete genes associated with the meQTL targets.

RESPONSE:

We have added a new supplementary table (Supplementary Table 1 in revised manuscript) that now includes the # of discrete genes among meQTLs, the # of discrete genes among meQTL

targets, and the # of non-overlapping genes among meQTL targets (i.e., were not implicated in the SNP list). We refer to this table in the results:

“In all tissues, meQTL targets (CpG sites controlled by meQTLs) implicate additional genes that are not accounted for by their corresponding meQTLs (Supplementary Table 1).”

2. Table 2: Are the p-values for the ASD-associated SNPs from the PGC study nominal or corrected for multiple testing? If nominal, can the authors justify the use of relatively liberal p-values (i.e., 1e-03 and 1e-004)?

RESPONSE:

These are nominal. Use of these more liberal thresholds is commonplace in other meQTL/eQTL enrichment analyses using GWAS results for psychiatric disorders^{3,5}, as well as in studies using polygenic risk scores⁸⁻¹⁰.

3. Gamazon, E. R. et al. Enrichment of cis-regulatory gene expression SNPs and methylation quantitative trait loci among bipolar disorder susceptibility variants. *Mol. Psychiatry* **18**, 340–346 (2013).
5. van Eijk, K. R. et al. Identification of schizophrenia-associated loci by combining DNA methylation and gene expression data from whole blood. *Eur. J. Hum. Genet. EJHG* **23**, 1106–1110 (2015).
8. Stringer, S., Kahn, R. S., de Witte, L. D., Ophoff, R. A. & Derks, E. M. Genetic liability for schizophrenia predicts risk of immune disorders. *Schizophr. Res.* **159**, 347–352 (2014).
9. Escott-Price, V. et al. Polygenic risk of Parkinson disease is correlated with disease age at onset. *Ann. Neurol.* **77**, 582–591 (2015).
10. French, L. et al. Early Cannabis Use, Polygenic Risk Score for Schizophrenia and Brain Maturation in Adolescence. *JAMA Psychiatry* **72**, 1002–1011 (2015).

3. With respect to the gene ontology analyses presented in Table 3, it would be much more informative to also present the lists of genes used for the GO analyses for each tissue.

RESPONSE:

The genes mapping to meQTL targets of ASD-related SNPs, for each tissue type, are listed in Supplementary Data 6. While we believe that the list of genes mapping to meQTL targets generally would be too lengthy to include in Table 3, we have provided a full list of peripheral blood (Supplementary Data 1) and cord blood (Supplementary Data 2) meQTL/meQTL targets, which can be used for gene lookup if needed.

4. Related to point 3, how do the genes implicated by the ASD-SNP associated meQTLs in this study compare with those identified by other methylation studies of ASD?

RESPONSE:

We agree that investigating ASD meQTL targets for overlap with CpGs identified from ASD epigenome-wide association studies would be of interest. However, we emphasize pathways in this report, rather than a focus on specific genes, for multiple reasons (see first response to

Reviewer #5). Further, there are few public reports of specific DNAm differences associated with ASD to date. The few that do exist, including ours from brain¹¹, are imprecise and heterogeneous with respect to tissue, platform, sample size, etc. Therefore, we did not attempt a direct overlap analysis. Instead, we do point out that the immune pathways implicated in our study by ASD meQTL targets are concordant with several ASD gene expression studies^{12–15} across blood and brain tissues and one ASD methylation study in brain¹⁶. Please see the section in the discussion beginning “Among CpG sites that are targets...”.

11. Ladd-Acosta, C. et al. Common DNA methylation alterations in multiple brain regions in autism. *Mol. Psychiatry* **19**, 862–871 (2014).

12. Gupta, S. et al. Transcriptome analysis reveals dysregulation of innate immune response genes and neuronal activity-dependent genes in autism. *Nat. Commun.* **5**, 5748 (2014).

13. Voineagu, I. et al. Transcriptomic analysis of autistic brain reveals convergent molecular pathology. *Nature* **474**, 380–384 (2011).

14. Kong, S. W. et al. Peripheral blood gene expression signature differentiates children with autism from unaffected siblings. *Neurogenetics* **14**, 143–152 (2013).

15. Jalbrzikowski, M. et al. Transcriptome Profiling of Peripheral Blood in 22q11.2 Deletion Syndrome Reveals Functional Pathways Related to Psychosis and Autism Spectrum Disorder. *PLoS One* **10**, e0132542 (2015).

16. Nardone, S. et al. DNA methylation analysis of the autistic brain reveals multiple dysregulated biological pathways. *Transl. Psychiatry* **4**, e433 (2014).

5. Fig. 1: What is/are the relationship(s) between the genes shown in Figure 1 and genes already associated with ASD?

RESPONSE:

We have modified Figure 1 to improve clarity and modified the legend accordingly. Please also see response to reviewer #4 above. Briefly, dark lines in Figure 1 show meQTL associations between ASD-associated PGC SNPs and their specific CpG targets.

6. Discuss the significance/meaning of the greatest number of ASD-SNP associated meQTLs in peripheral blood relative to cord blood and fetal brain.

RESPONSE:

This is due to the much greater sample size available for peripheral blood meQTL discovery. We tried to address the difference in detectable effects through our data-driven approach to identify meQTLs (see Methods section ‘meQTL identification parameters’), but the sheer sample size difference necessarily results in a greater number of significant SNP-CpG associations. Our approach does make the enrichment and downstream pathway analyses comparable across tissues despite the different starting list sizes. Moreover, we performed down sampling to account for this issue in the regulatory feature enrichment analyses (see Methods section ‘Regulatory feature characterization of meQTL targets’). We have added the following underline

phrase to a sentence from the Methods section 'meQTL identification parameters' to emphasize the differing sample sizes:

"Our available sample sizes (and thus statistical power, at fixed effect size) for the joint DNAm and genotype data differed for peripheral blood (n = 339) and cord blood (n = 121) analyses after limiting both to samples of European descent identified via principle components analysis of SNP data."

7. Fig. 2: The enrichment of meQTL targets in DNase hypersensitivity sites shown for PGC meQTL targets vs. non-PGC meQTL targets is very interesting. What is the analogous DHS enrichment data for ASD-meQTL targets vs. the non-PCG meQTL targets?

RESPONSE:

We agree than an ASD-specific meQTL target enrichment analysis would be of interest, but please see the response to Reviewer #2 regarding regulatory feature analysis. Our primary goal in this specific features analysis was to characterize *cross-tissue* meQTL targets with respect to their functional propensity, because we could not find such information yet in the literature. Doing this analysis in the context of a broader psychiatric phenotype allowed for a greater number of GWAS SNPs to be used and therefore a greater number of relevant meQTL targets. This in turn ensured an adequate number of meQTL targets that overlapped across tissues to enable a meaningful enrichment analysis. While in the future we hope to ask these same regulatory feature enrichment questions in the context of ASD specifically, current data identifying ASD meQTL targets precludes a well-powered analysis. Our results are still relevant to ASD in light of known cross-disorder GWAS consistency.

Minor points:

8. Define what is meant by "meQTL targets"

RESPONSE: Thank you for this suggestion. We have added clarifying language at numerous points in the manuscript. For example, in the Results section 'Identifying meQTLs', we have added the underlined phrase:

"In all tissues, meQTL targets (CpG sites controlled by meQTLs) implicate additional genes that are not accounted for by their corresponding meQTLs (Supplementary Table 1)."

9. Page 14, 1st paragraph, line 6: "The 232 mothers with a subsequent sibling..." should be "...subsequent child..."

RESPONSE:

We have corrected this.

10. What were the total numbers of fetal brain and lung tissues analyzed for meQTLs?

RESPONSE:

Fetal brain results were derived from a previous study of 166 samples. Lung results were derived from a previous study of 210 samples. These numbers are included in Table 1.

11. Page 18, second paragraph, last sentence: "...to be comparable." should be "...to be compared."

RESPONSE: We have revised this sentence for clarity:

"Thus, the ideal combination of these parameters should differ between the two study populations for them to be comparable."

12. Page 23, last paragraph, line 4: "...were downloaded in from GEO..." Remove "in".

13. Footnote to Table 3: "NA shown for terms that did appear..." I think the authors meant "...did not appear..."

RESPONSE:

We have corrected these.

REVIEWERS' COMMENTS:

Reviewer #2 (Remarks to the Author):

The authors have addressed my concerns adequately and I am in support for the manuscript's publication.

Reviewer #4 (Remarks to the Author):

Generally the manuscript reads well and presents an interesting approach for a combined analysis of methylation and SNPs for cross-tissue associations relevant to ASD.

Reviewer #5 (Remarks to the Author):

As mentioned previously, this study describes a novel way to integrate genome-wide methylation and genetic data to reveal additional genes that may contribute to ASD. Although still a "dense" manuscript, the authors have addressed all of my questions/comments to my satisfaction.